# Molecular Marker-Assisted Selection of *ABCG2*, *CD44*, *SPP1* Genes Contribute to Milk Production Traits of Chinese Holstein

**DOI:** 10.3390/ani13010089

**Published:** 2022-12-26

**Authors:** Yujia Sun, Xinyi Wu, Yaoyao Ma, Dingding Liu, Xubin Lu, Tianqi Zhao, Zhangping Yang

**Affiliations:** 1Joint International Research Laboratory of Agriculture and Agri-Product Safety, The Ministry of Education of China, Institutes of Agricultural Science and Technology Development, Yangzhou University, Yangzhou 225009, China; 2Key Laboratory of Animal Genetics & Breeding and Molecular Design of Jiangsu Province, Yangzhou University, Yangzhou 225009, China

**Keywords:** *ABCG2*, *CD44*, *SPP1*, association analysis, SNP loci, production performance, Chinese Holstein cattle

## Abstract

**Simple Summary:**

Chinese Holstein cattle are the main breed of dairy cows in China. This work performed an association analysis between polymorphisms of three candidate genes (*ABCG2*, *CD44*, *SPP1*) and milk production performance on Chinese Holstein cattle. The results showed that the different genotypes of these 10 SNPs from the *ABCG2*, *CD44*, and *SPP1* genes significantly affected the milk production performance of Chinese Holstein cattle in terms of milk yield, milk fat percentage, milk protein percentage, somatic cell score, and urea nitrogen content. The *ABCG2*, *CD44*, and *SPP1* genes could be selected for marker-assisted selection, which is of great significance for future precise molecular breeding.

**Abstract:**

Based on our results of genome-wide association analysis, we performed gene ontology (GO) annotation and Kyoto Encyclopedia of Genes and Genomes (KEGG) pathway enrichment analysis; three candidate genes (*ABCG2*, *CD44*, *SPP1*) were screened in this study for SNPs association analysis with production traits in 999 Holstein cattle. In this research, flight mass spectrometry genotyping was used to detect the polymorphism of SNP seats. It was shown that four, four, and two single nucleotide polymorphisms (SNP) loci were detected for the *ABCG2*, *CD44*, and *SPP1* genes, respectively, and the different genotypes of these 10 SNPs significantly affected the milk production performance of Chinese Holstein cattle in terms of milk yield, milk fat percentage, milk protein percentage, somatic cell score, and urea nitrogen content. Among them, *ABCG2*-G.80952G > T locus, *ABCG2*-G.120017G > A locus and *CD44*-G.2294G > C locus had significant effects on somatic cell score (*p* < 0.01). Cows with GG genotypes at *ABCG2*-G.80952G > T locus, AA and GG genotypes at *ABCG2*-G.120017G > A locus, and GG genotypes at *CD44*-G.2294G > C locus had lower somatic cell scores. The present study elucidated that *ABCG2*, *CD44,* and *SPP1* could be selected for marker-assisted selection and will benefit for future precise molecular breeding.

## 1. Introduction

The performance of dairy cows mainly refers to milk production traits. Milk-producing traits have always been the more concerned quantitative economic traits in dairy farming, which are jointly controlled by micro-efficiency polygenes and are susceptible to environmental influences [1]. Due to the long generation interval of dairy cows, the selection of milk producing traits of dairy cows is slow in conventional breeding work. Therefore, molecular marker-assisted breeding is a new situation in the current breeding work. Single nucleotide polymorphisms (SNP) have been widely used in DNA molecular genetic markers, and the prospects are very promising [2].

SNP is the third generation of molecular markers after restriction fragment length polymorphism (RFLP) and microsatellite polymorphism (MPP) [3]. SNP refers to DNA sequence polymorphism caused by single nucleotide variation in the genome, including the insertion, deletion, transversion, and transition of single bases, with an allele variation frequency above 1%. SNP mutations may cause changes in genetic codons, and different genetic codons encode different proteins, and protein changes directly affect biological diversity [4,5]. Therefore, the change in the SNP mutation site may cause a change in individual traits, such as growth, reproduction, milk production, and other traits. Through the continuous reproduction of the population, the good characters are preserved, and the unsuitable characters are eliminated. If it is applied to the breeding of livestock and poultry, the selection and breeding of excellent breeds can be accelerated. More and more studies have found that SNPs are important to the growth, production, reproduction, disease, and other traits of cattle [6].

In our previous work, we used the recorded data of 86,281 test days of 8580 Holstein cows in Jiangsu, and we used the random regression measurement day model to estimate the genetic parameters of Holstein somatic cell scores (SCS). At the same time, the adjusted phenotype was used to perform a genome-wide association analysis (GWAS) of dairy cow SCS using the FarmCPU method to screen for significant SNPs and candidate genes [7,8].

Based on the results of GWAS, we performed gene ontology (GO) annotation and Kyoto Encyclopedia of Genes and Genomes (KEGG) pathway enrichment analysis; three candidate genes (*ABCG2*, *CD44*, *SPP1*) were selected in this study for SNPs association analysis with production traits in Holstein cattle. The present study elucidated that *ABCG2*, *CD44,* and *SPP1* closely related to dairy cattle performance were selected for the SNPs screening, which contributed to the marker-assisted selection and is of great significance for future precise molecular breeding.

## 2. Materials and Methods

### 2.1. Sample Collection

The experimental population of this study was from four large-scale dairy farms in Jiangsu, China. In this study, SNP microarray samples were collected from cows for genetic parameter evaluation and association analysis. SNP chips were used to collect samples of hair follicles from cows, and hair follicle samples from 999 cows in four farms were collected from 8580 cows that passed the quality control, these experimental cows were of the same age (24 months) and the same parity (parity 2). The collected samples were sent to Neogen Biotechnology Co. Ltd (Lansing, MI, USA), and DNA was extracted and genotyped from the hair follicles of 999 cows.

### 2.2. Enrichment Analysis and DNA Detection

The database (DAVID) online software (https://david.ncifcrf.gov, accessed on 4 November 2022) and the bovine reference genome ARS-UCD1.2(ftp://hgdownload.soe.ucsc.edu/golden path/bosTau9/, accessed on 10 September 2022) in the UCSC database were used, and other online tools have carried out the GO (https://david.ncifcrf.gov, accessed on 4 November 2022) annotation and KEGG (https://www.kegg.jp/, accessed on 4 November 2022) pathway enrichment analysis of candidate gene functions. DNA purity and concentration were tested as follows: using NanoDroP1000 spectrophotometer can directly detect the ratio of DNA OD260/OD280, OD260/OD230 and the concentration of DNA, using the following formula to calculate the concentration of DNA: DNA concentration (ug/mL) = 50 × OD260, when the ratio of OD260/OD280 is between 1.7–1.9, it indicates that the concentration of DNA is relatively ideal.

### 2.3. PCR Amplification, and Commercial Sequences

Based on the sequences of the bovine *ABCG2*, *SPP1,* and *CD44* genes published on NCBI, primers were designed by software in the CDS region and part of the intron region of the *ABCG2*, *SPP1,* and *CD44* genes, respectively. A total of 7 pairs of primers were involved in this test, and the details of the primers were shown in Table 1.

### 2.4. Screening of SNP Mutation Sites

20 DNA samples were randomly selected for each pair of primers for PCR amplification, and the successfully amplified products were sent to Qingke Biological Company (Nanjing, China) for sequencing. The sequencing results were compared with Vector NTI Advance 11 software to find out the mutation sites and their specific positions. The amplification procedure was as follows: pre-denaturation at 94 °C for 5 min, denaturation at 94 °C for 30 s, renaturation at 68 °C for 30 s, and extension at 72 °C for 2 min 10 s, 17 cycles. Denaturation at 94 °C for 30 s, renaturation at 51 °C for 30 s, and extension at 72 °C for 2 min 10 s, 20 cycles. The final extension was at 72 °C for 10 min and stored at 4 °C.

### 2.5. Fractal Detection of SNPs

The screened mutation loci were determined by matrix-assisted laser desorption ionization time-of-flight mass spectrometry (MALDI-TOF-MS) for typing detection. The genotyping detection system adopts the time-of-flight mass spectrometry biochip system (MassARRAY® MALDI-TOF System) developed by Sequenom, Inc. (San Diego, CA, USA). This system has been widely promoted and used in the Human Genome Hapmap Project.

### 2.6. Statistical Analysis

The Hardy-Weinberg equilibrium test (HWE), genotype and allele frequency analysis, and linkage disequilibrium (LD, as measured by D′ and r^2^ ) analysis were performed for SNP sites after *ABCG2*, *SPP1,* and *CD44* genotyping by SHEsis software (http://analysis2.bio-x.cn/myAnalysis.php, accessed on 4 November 2022) [9]. Multi-factor ANOVA with SPSS (Ver. 18.0) software and general linear models were used to perform the association analysis between the genotype and production performance of *ABCG2*, *SPP1*, and *CD44* gene SNPs loci. The reduced linear model excluded fixed effects of age and paternity. The specific models were as follows:*Y_ijklmnop_* = *μ* + *C_i_* + *M_j_* + *P_k_* + *J_l_* + *N_m_* + *G_n_* + *F_o_* + *e_ijklmnop_*
where *Y_ijklmnop_* is the observed value of the trait, *μ* is the population mean, and *C_i_* is the fixed effect of calving season. *M_j_* is the fixed effect of the lactation stage, *P_k_* is the fixed effect of parity, *J_l_* is the fixed effect of measured season, *N_m_* is the fixed effect of the test year, *G_n_* is the fixed effect of genotype, *F_o_* is the cattle field effect, and *e_ijklmnop_* is the random error. Multiple comparisons between genotypes were obtained using Duncan’s method.

## 3. Results

### 3.1. Functional Annotation and Signal Pathway Analysis of ABCG2, CD44 and SPP1 Genes

The profiles of three genes were in the biological process (BP), cell component (CC), and molecular function (MF) categories by GO analysis, respectively. The BP analysis showed that the three genes were classified into the cellular process, localization, metabolic process, response to stimulus, and single-organism process. The CC annotation revealed that the three genes were involved in the cell, cell part, and organelle. The MF classification of genes suggested that three genes were involved in binding. GO annotation analysis elucidated that the three genes participated in the regulation of production performance by multiple indispensable activities (Figure 1A).

Furthermore, we performed KEGG pathway enrichment analysis to explore the most active pathways of these genes; the results revealed that these genes were related to signaling molecules and interactions, as well as the immune system and infectious diseases, which are partially but not completely involved (Figure 1B).

### 3.2. Screening of ABCG2, CD44 and SPP1 Genes SNP Mutation Loci

By comparing the sequencing results with the three gene sequences, four SNP loci were found, respectively, in *ABCG2* and *CD44*: *ABCG2*-g.57261A > G, *ABCG2*-g.80952G > T, *ABCG2*-g.94683A > G, *ABCG2*-g.120017G > A, *CD44*-g.2263A > G, *CD44*-g.2294G > C, *CD44*-g.86895A > G, *CD44*-g.86978G > A. *SPP1* was found two SNP loci *SPP1*-g.50265G > A and *SPP1*-g.50315C > T. Figure 2A–D, Figure 3A–D, and Figure 4A,B show the sequencing peaks of each SNP locus of *ACG2*, *CD44,* and *SPP1*, respectively.

### 3.3. LD Analysis of SNPS in ABCG2, CD44 and SPP1 Genes

LD analysis was performed on the SNP loci of the *ABCG2*, *CD44,* and *SPP1* genes with SHEsis software, and the results were obtained by D′ value and r^2^ value. The D′ > 0.7 and r^2^ > 1/3 might suggest a sufficiently strong LD to be available for plotting [10]. All the D’ and r^2^ values among the different SNP loci of the *ABCG2*, *CD44,* and *SPP1* genes listed in Table 2. The *ABCG2* gene *ABCG2*-g.57261A > G and *ABCG2*-g.94683A > G, *ABCG2*-g.120017G > A, *ABCG2*-g.80952G > T and *ABCG2*-g.120017G > A are in a highly linked state (D′ > 0.7, r^2^ > 1/3), respectively, and the linkage degree between the remaining loci is relatively low (Figure 5A). The *CD44* gene *CD44*-g.2294G > C and *CD44*-g.86978G > A are in a highly linked state (D′ > 0.7, r^2^ > 1/3), and there is no high degree of linkage between other loci chain relationship (Figure 5B). There is no high linkage relationship between the two SNPs of the *SPP1* gene (D′ > 0.7, r^2^ < 1/3) (Figure 5C).

### 3.4. Genotype Frequency and Allele Frequency of SNP Loci in ABCG2, CD44 and SPP1 Genes

As shown in Table 2, among the four SNP loci of *ABCG2* gene, *ABCG2*-g.57261A > G and *ABCG2*-g.80952G > T were located in intron 1, *ABCG2*-g.94683A > G in intron 5, and *ABCG2*-g.120017G > A in intron 13. *ABCG2* gene has three genotypes for each SNP locus The dominant genotypes of *ABCG2*-g.57261A > G, *ABCG2*-g.80952G > T, *ABCG2*-g.94683A > G and *ABCG2*-g.120017G > A were GG, GT, AG, and AG, respectively. After χ2 test, the 4 SNPs of *ABCG2* gene all reached HWE (*p* > 0.05). Among the four SNPs of *CD44* gene, *CD44*-G.2263a > G and *CD44*-G.2294G > C were located in intron 2, and *CD44*-G.86895A > G and *CD44*-G.86978G > A were located in exon 17. *CD44*-g.2294G > C, *CD44*-g.2263A > G, *CD44*-g.86895A > G and *CD44*-g.86978G > A also had three genotypes, respectively. The dominant gene of *CD44*-g.2263A > G, *CD44*-g.86895A > G and *CD44*-g.86978G > A was AA, and the *CD44*-g.2294G > C was CG. Similarly, the two SNP loci of *SPP1* also had three genotypes, respectively, and the dominant genotypes were GG and CT. After χ2 test, all SNP loci of the three genes were in HWE (*p* > 0.05) (Table 3).

### 3.5. Correlation Analysis between Gene SNP Sites and Production Performances

#### 3.5.1. Association Analysis of Four SNP Loci in *ABCG2* Gene and Production Traits

The correlation analysis results of the four SNP loci of the BCG2 gene with different genotypes of the Holstein cattle somatic cell score, urea nitrogen in milk, daily milk yield, milk protein rate, and milk fat rate are shown in Table 4. The effect of *ABCG2*-g.57261A > G locus on milk fat rate reached a significant level (*p* < 0.01). The milk fat rate of GG- type cows was the highest, and the AA genotype was the lowest. The *ABCG2*-g.94683A > G locus had a highly significant effect (*p* < 0.01) on the urea nitrogen content in milk, which was significantly higher in the milk of AA-type cows than in AG and GG types. The *ABCG2*-g.80952G > T locus had a highly significant (*p* < 0.01) effect on milk yield, milk protein rate, somatic cell score, and urea nitrogen in cows, with TT-type cows having significantly lower measured daily milk yield than GG and GT types, while TT-type cows had significantly higher milk protein rate, somatic cell score, and urea nitrogen than GG and GT types. *ABCG2*-g.120017G > A had significant effects on milk yield, milk protein percentage, somatic cell score, and urea nitrogen (*p* < 0.01) and had significant effects on milk fat percentage (*p* < 0.05). The milk protein percentage, milk fat percentage, somatic cell score, and urea nitrogen of AG cows were significantly higher than those of AA and GG cows. A square visual diagram of *ABCG2* gene difference expression is shown in Figure 6.

#### 3.5.2. Association Analysis of 4 SNP Loci of *CD44* Gene and Production Traits

The association analysis results of different genotypes of the 4 SNP loci of *CD44* gene with somatic cell score, milk urea nitrogen, measured daily milk yield, milk protein percentage, and milk fat percentage of Holstein cattle are shown in Table 5. The effect of *CD44*-g.2263A > G locus on milk urea nitrogen reached a very significant level (*p* < 0.01), and the urea nitrogen content of AA cows was significantly higher than that of AG and GG genotypes. The *CD44*-g.86978G > A locus had a very significant effect on the milk fat rate of dairy cows (*p* < 0.01), and the AA type was the highest. *CD44*-g.2294G > C locus had extremely significant effects on SCS and milk urea nitrogen contents (*p* < 0.01) and had significant effects on milk fat percentage (*p* < 0.05). CG-type cows had a higher milk fat percentage and milk urea nitrogen content, which were significantly higher than CC-type cows and CG-type cows; CC-type cows and CG-type cows had higher SCS. The *CD44*-G.86895A > G locus had significant effects on milk yield, milk fat rate, protein rate, and milk urea nitrogen content of dairy cows (*p* < 0.01). The milk yield, milk fat percentage, and milk urea nitrogen content of GG dairy cows were the highest, as they were significantly higher than those of other genotypes, and the protein percentage of AA and AG cows was the highest. A square visual diagram of *CD44* gene difference expression is shown in Figure 7.

#### 3.5.3. Association Analysis of Two SNP Loci in *SPP1* Gene and Production Traits

The correlation analysis results of the two SNP loci of the *SPP1* gene between different genotypes and Holstein cattle somatic cell score, urea nitrogen in milk, daily milk yield, milk protein rate, and milk fat rate are shown in Table 6. The *SPP1*-g.50265G > A locus had a significant level of effect on milk yield *p* < 0.05) and had a highly significant effect on milk urea nitrogen content (*p* < 0.01). The milk yield and milk urea nitrogen content of the AG and GG cows were significantly higher than those of the AA cows. The *SPP1*-g.50315 C > T locus had a significant effect on the milk fat rate of dairy cows (*p* < 0.05), and the CT-type and TT-type were significantly higher than the CC-type. A square visual diagram of *SPP1* gene difference expression is shown in Figure 8.

## 4. Discussion

Previous studies analyzed the association between *ABCG2* gene polymorphism and milk yield and milk composition of Kalanfris cattle and found that the three genotypes of SNP on exon 14 had significant effects on the milk fat percentage of cows [11]. Tantia identified the quantitative trait loci of Indian cattle (Bos indicus) and river cattle (Bubalus bubalis) and found that the allele of *ABCG2* had a significant impact on high milk fat yield, high fat, and protein percentage [12]. In this study, four polymorphisms were found in the *ABCG2* gene, and all these polymorphisms were located in the intronic region. Existing studies have shown that introns can generate splicing signals through their own sequence properties to affect transcript splicing, thereby affecting protein sequences and individual phenotypes [13]. The results of this study showed that *ABCG2*-G.57261A > G locus had a significant effect on the milk fat percentage of dairy cows, and GG type was the highest. Milk urea nitrogen content in the AA genotype of *ABCG2*-G.94683A > G locus was significantly higher than that in the AG and GG genotypes. Milk protein percentage, somatic cell score and urea nitrogen of cows carrying *ABCG2*-G.80952G > T TT genotype were significantly higher than those of the GG and GT genotypes. Milk protein percentage, milk fat percentage, somatic cell score, and urea nitrogen were significantly higher in cows carrying the AG genotype at the *ABCG2*-g.120017G > A locus than in the AA and GG types. The effects of *ABCG2*-G.80952G > T and *ABCG2*-G.120017G > A on production traits of dairy cows showed similar trends, probably because of the high linkage relationship between them. The results of this study show that *ABCG2* polymorphisms have significant effects on dairy cows’ milk fat rate, milk yield, milk protein rate, and other production traits, which are basically consistent with previous studies. In the selection of dairy cows, the selection of *ABCG2*-g.57261A > G site GG type and *ABCG2*-g.120017G > A site AG type dairy cows should be increased in view of improving milk fat rate. In order to improve milk production and milk protein rate of dairy cows, the selection of TT-type cows at *ABCG2*-g.80952G > T site and AG-type cows at *ABCG2*-g.120017G > A site can be strengthened. However, the SCS of cows with these two genotypes are also relatively high, which may increase the burden of udder function with the increase in milk production, aggravate the shedding of mammary epithelial cells, and increase the SCS of cows. Olsen [14] found an SNP mutation in the *ABCG2* gene in exon 9, resulting in a mutation of the amino acid from tyrosine to cysteine, but the differences in the five lactation performances among the three genotypes were not significant. In the selection of dairy cow genotypes, it is necessary to comprehensively select the SNPs genotypes corresponding to the target traits, so we are required to use different methods to find more SNP loci that affect dairy cow traits.

Most of the existing studies on *CD44* have focused on disease treatment. *CD44* gene rs13347 polymorphism is significantly associated with elevated cancer risk in Asians [15]. Deletion of the *CD44* gene has been reported to reduce proinflammatory cytokines [16], and treatment of affected mice with *CD44* antibodies reduced inflammation [17]. Previous studies on *CD44* in cattle have shown that *CD44* is related to dairy production traits. Jiang [18] believed that the *CD44* gene could affect the synthesis of triglyceride in bovine mammary epithelial cells, thereby affecting the lipid content in milk. A total of 4 SNP loci were found in this study, *CD44*-g.2263A > G and *CD44*-g.2294G > C were located in the intron region, *CD44*-G.86895A > G and *CD44*-G.86978G > A were located in the exon region, the *CD44*-g.86895A > G site is a nonsense mutation, and the *CD44*-g.86978G > A site is a missense mutation. Association analysis found that the milk urea nitrogen content of cows with *CD44*-g.2263A > G locus AA genotype was significantly higher than that of AG and GG genotypes. The *CD44*-g.86978G > A locus had a significant effect on milk fat percentage in cows, with the AA type having the highest. Cows with the *CD44*-G. 2294G > C CG genotype had a higher milk fat percentage and milk urea nitrogen content, which were significantly higher than those with CC and CG genotypes, while cows with CC and CG genotypes had higher SCS. Cows carrying the *CD44*-g.86895A > G locus GG genotype had the highest milk yield, milk fat percentage, and milk urea nitrogen content, which were significantly higher than other genotypes, while cows with AA and AG genotypes had the highest protein rate. From the results of this study, the genotypes of the three SNP loci have a significant impact on the milk fat rate of dairy cows, which proves that *CD44* is related to the lipid metabolism of dairy cows. This result is consistent with previous studies. In order to improve milk fat percentage, the selection of dairy cows carrying *CD44*-G.86978G > A AA genotype, *CD44*-G.2294G > C CG genotype and *CD44*-G.86895A > G GG genotype should be strengthened. There are more and more studies on *SPP1* gene polymorphism, which provides a powerful reference for disease treatment and improving livestock and poultry production.

Maria [19] conducted an association analysis of the *SPP1* gene polymorphism in Salda sheep and found that rs161844011 locus in exon 7 was associated with SCS. Liang [20] found that people carrying the CC genotype and C allele of the rs11730582 locus of the *SPP1* gene had a reduced risk of developing breast cancer. Furthermore, *SPP1* gene polymorphism was significantly correlated with lactation persistence of dairy cows, and G was its dominant allele [21]. Mello analyzed the association between *SPP1* gene polymorphism and milk yield of Girorando cows and found that 305-day cows carrying the T allele at position g.8514C > T had higher milk yield, but it did not reach a significant level [22]. The SNPs of the two *SPP1* genes obtained in this study were located in the intronic region. The milk yield and milk urea nitrogen content of the AG and GG genotypes of *SPP1*-G. 50265G > A were significantly higher than those of the AA genotypes. *SPP1*-g.50315 C > T loci CT and TT genotypes had significantly higher milk fat percentages than the CC types. The selection of cows carrying the G allele at the *SPP1*-G.50265G > A locus can be strengthened in future dairy breeding to improve milk yield. From the perspective of improving the milk fat rate of dairy cows, the selection of dairy cows carrying the *SPP1*-g.50315 C > T locus T allele can be strengthened.

## 5. Conclusions

This work performed GO annotation and KEGG pathway enrichment analysis on the *ABCG2*, *CD44* and *SPP1* genes and analyzed the association between the *ABCG2*, *CD44,* and *SPP1* genes and Chinese Holstein production performance. The milk yield, milk fat rate, milk protein rate, somatic cell score, and blood urea nitrogen content of Holstein cattle with different genotypes of these 10 SNP loci were different. Among them, *ABCG2*-G.80952G > T locus, *ABCG2*-G.120017G > A locus and *CD44*-G.2294G > C locus had significant effects on the somatic cell score (*p* < 0.01). The present study elucidated that *ABCG2*, *CD44* and *SPP1* could be selected for marker-assisted selection and will benefit future precise molecular breeding.

## Figures and Tables

**Figure 1 animals-13-00089-f001:**
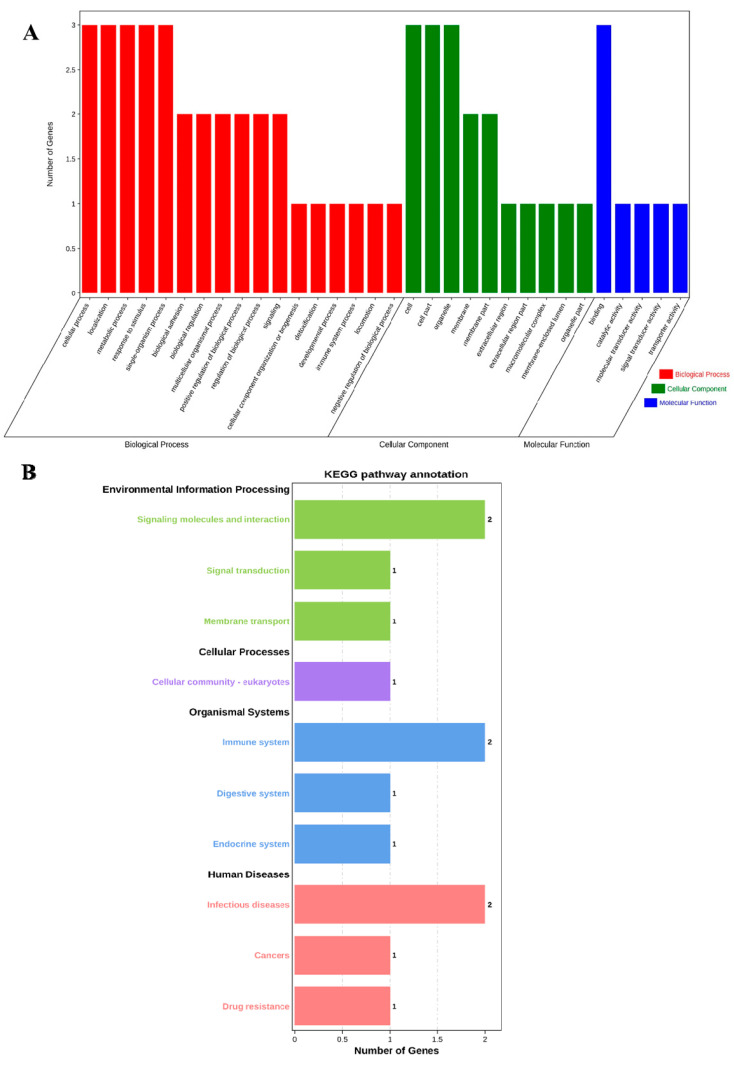
GO annotation and KEGG pathway enrichment of *ABCG2*, *CD44* and *SPP1* genes. (**A**) GO annotation results of three genes in BP, CC and MF. (**B**) Distribution of enriched KEGG pathway.

**Figure 2 animals-13-00089-f002:**
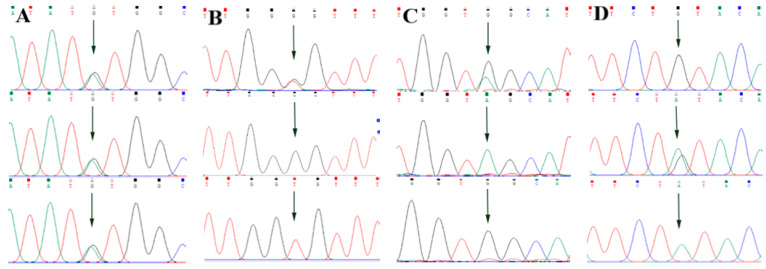
Sequencing peak map of four SNPs in *ABCG2* gene sequence. The sequence map at *ABCG2*-g.57261A > G site for AG, AA and GG (**A**), The sequence map at *ABCG2*-g.80952G > T site for GT, GG and TT (**B**), The sequence map at *ABCG2*-g.94683A > G site for AG, AA and GG (**C**), The sequence map at *ABCG2*-g.120017G > A site for AG, GG and AA (**D**). The arrow indicates the location of the mutation sites.

**Figure 3 animals-13-00089-f003:**
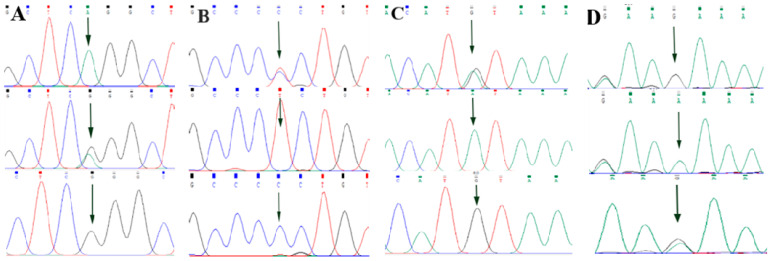
Sequencing peak map of four SNPs in *CD44* gene sequence. The sequence map at *CD44*-g.2263A > G site for AG, AA and GG (**A**), The sequence map at *CD44*-g.2294G > C site for CG, GG and CC (**B**), The sequence map at *CD44*-g.86895A > G site for AG, AA and GG (**C**), The sequence map at *CD44*-g.86978G > A site for AG, AA and GG (**D**). The arrow indicates the location of the mutation sites.

**Figure 4 animals-13-00089-f004:**
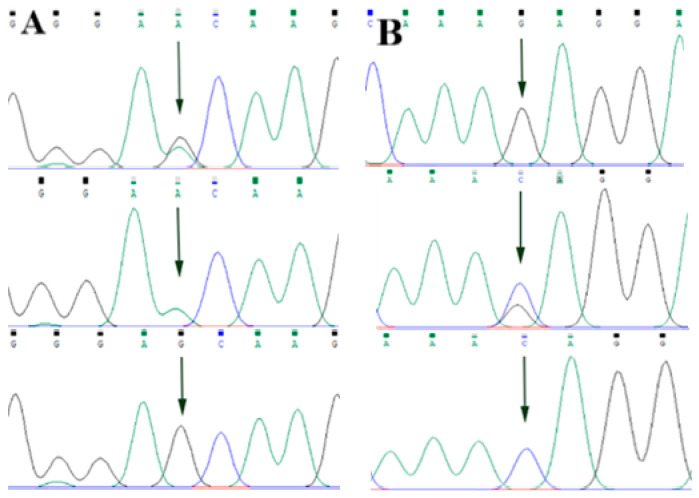
Sequencing peak map of two SNPs in *SPP1* gene sequence. The sequence map at *SPP1*-g.50265G > A site for GA, GG and AA (**A**), The sequence map at *SPP1*-g.50315C > T site for CT, TT and CC (**B**). The arrow indicates the location of the mutation sites.

**Figure 5 animals-13-00089-f005:**
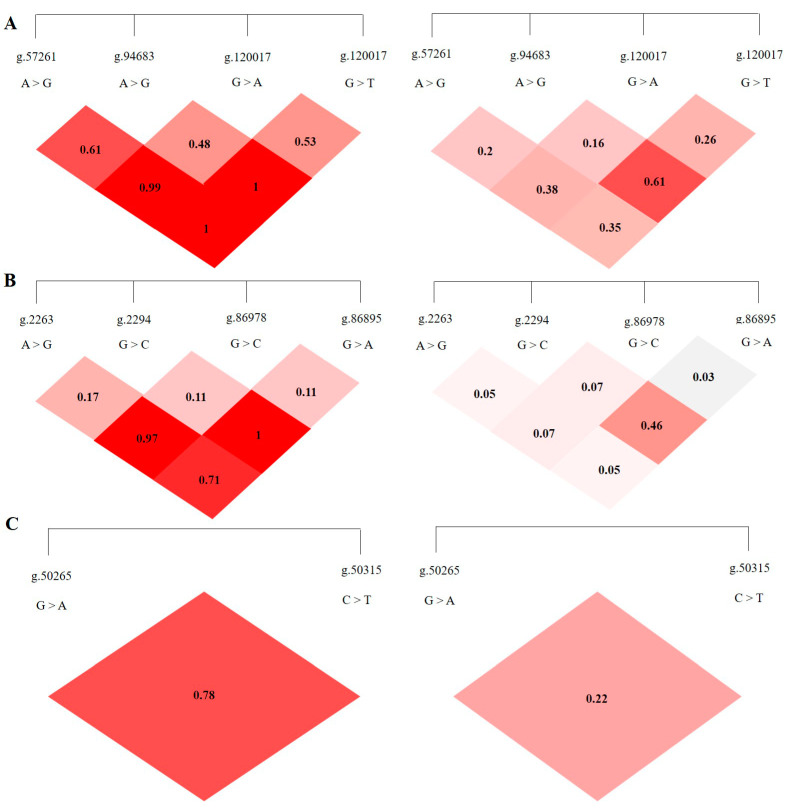
The linkage map of SNPs loci of *ABCG2*, *CD44*, *SPP1* genes. LD value D′ > 0.7, r^2^ > 1/3 indicates relatively high linkage strength. (**A**) The LD map of 4 SNPs loci of *ABCG2* gene. (**B**) The LD map of 4 SNPs loci of *CD44* gene. (**C**) The LD map of 2 SNPs loci of *SPP1* gene.

**Figure 6 animals-13-00089-f006:**
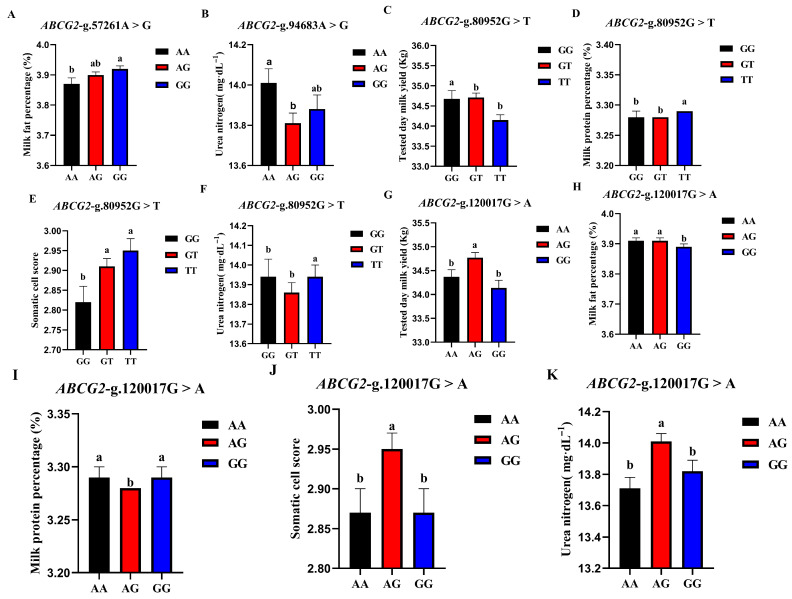
Effect of different genotypes in *ABCG2* four SNP loci on production traits in Chinese Holstein cattle. (**A**) The milk fat percentage of GG-type cows in the *ABCG2*-g.57261A > G locus was significantly higher than that of AA-type and AG-type cows. (**B**) *ABCG2*-g.94683A > G locus had significantly higher urea nitrogen content in the milk of AA type cows than AG and GG types. In *ABCG2*-g.80952G > T locus, the tested day milk yield (**C**) of TT type cows was significantly lower than that of GG and GT type cows; somatic cell score (**D**), milk protein percentage (**E**), and urea nitrogen (**F**) of TT type cows were significantly higher than GG and GT types. In *ABCG2*-g.120017G > A locus, AA type cows tested day milk yield (**G**) was significantly higher than that of AG and GG type cows, and the milk protein percentage (**H**), milk fat percentage (**I**), somatic cell score (**J**) and urea nitrogen (**K**) of AG type cows were significantly higher than those of AA and GG type cows. Data in the same column with different lowercase letters on the shoulder indicate significant differences (*p* < 0.05).

**Figure 7 animals-13-00089-f007:**
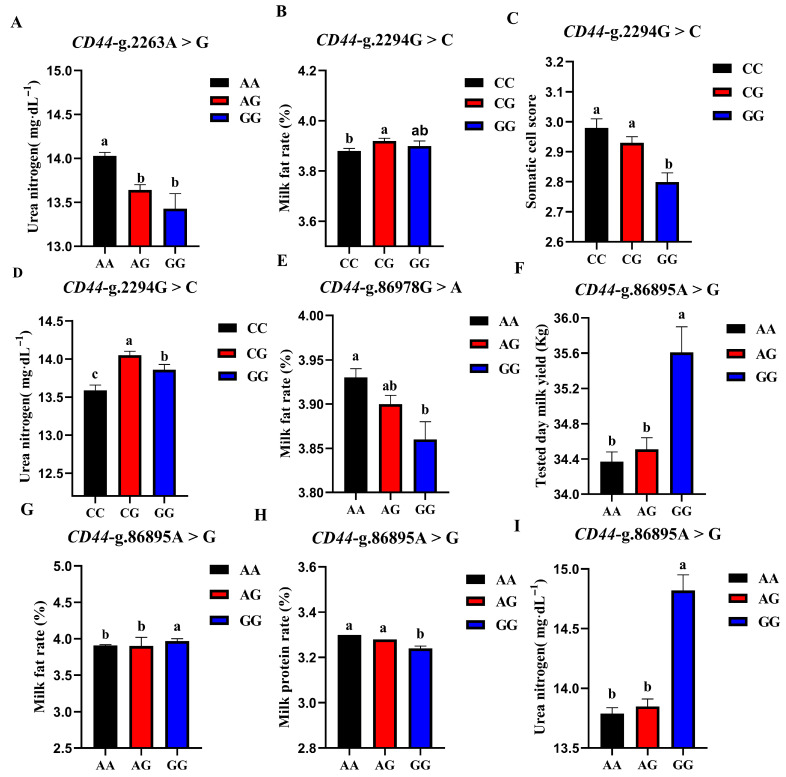
Effect of different genotypes in *CD44* four SNP loci on production traits in Chinese Holstein cattle. In *CD44*-g.2263A > G locus, AA type urea nitrogen content of was significantly higher than that of AG and GG genotypes (**A**). In *CD44*-g.2294G > C locus, milk fat percentage (**B**) and milk urea nitrogen content of CG-type (**D**) cows were significantly higher than CC and CG types. Higher SCS for CC- and CG-type cows (**C**). In *CD44*-g.86978G > A locus, Type AA cows have the highest milk fat percentage (**E**). In *CD44*-g.86895A > G locus, tested day milk yield (**F**), milk fat rate (**G**), and milk urea nitrogen content (**I**) of GG-type cows were significantly higher than those of AA, AG types; AA and AG cows had the highest protein rates (**H**). Data in the same column with different lowercase letters on the shoulder indicate significant differences (*p* < 0.05).

**Figure 8 animals-13-00089-f008:**
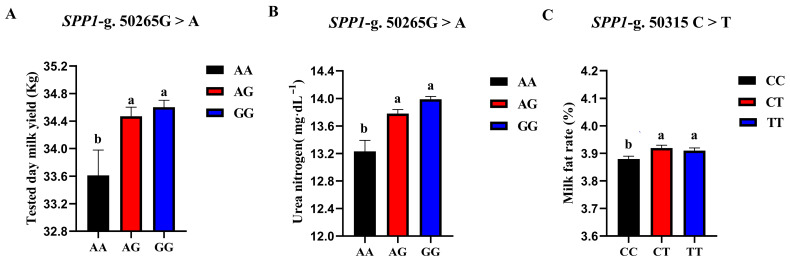
*SPP1*-g.50265G > A and *SPP1*-g.50315C > T loci on the productive performance of different genotypes. tested day milk yield (**A**) and milk urea nitrogen content (**C**) were significantly higher in AG and GG types of cows than in AA-type; CT and TT types of cows were significantly higher than the CC-type, milk fat rate (**B**) of CT and TT types were significantly higher than that of CC-type. Data in the same column with different lowercase letters on the shoulder indicate significant differences (*p* < 0.05).

**Table 1 animals-13-00089-t001:** Primer sequence and amplified fragment length of gene PCR amplification.

Primer	Sequences of Primer	Size of Production (bp)	Position of Production
P1 (*ABCG2*)	F: AAGGAGGAAAGGAGCCAGAG	460	57,031
	R: TGCTACCAGACACGAAATCG		
P2 (*ABCG2*)	F: TTGGATGATGATGACTTTGG	766	80,828
	R: GAACTTTCTCTCTGGCTACTG		
P3 (*ABCG2*)	F: TGCTTTCAACTTCTCTGCTC	392	94,583
	R: GTCCTTTTTCTTTCTCCTCC		
P4 (*ABCG2*)	F: TGGTTATATTGGGTGGTTGG	606	119,653
	R: ACTATGGGATGAGGTTCGTG		
P5 (*CD44*)	F: GCTTTGCTTCTGAGGATTCTG	590	1985
	R: TCGCTTCACTGCTCTTTACC		
P6 (*CD44*)	F: CCCGCTCCTCGAGTTTTCTG	324	86,737
	R: ATTGAGTCCGCTGGGCTTTC		
P7 (*SPP1*)	F: AATAAACCCTTTTCCCTCCC	495	50,063
	R: CCTTACAAATTGACCTTCCC		

**Table 2 animals-13-00089-t002:** The estimated values of LD analysis between mutation sites within the *ABCG2*, *CD44*, *SPP1* genes.

*ABCG2* Loci	g.57261A > G	g.80952G > T	g.94683A > G	g.120017G > A
g.57261A > G		D′ = 0.61	D′ = 0.99	D′ = 1.00
g.80952G > T	r^2^ = 0.20		D′ = 0.48	D′ = 1.00
g.94683A > G	r^2^ = 0.38	r^2^ = 0.16		D′ = 0.53
g.120017G > A	r^2^ = 0.35	r^2^ = 0.61	r^2^ = 0.26	
***CD44* Loci**	g.2263A > G	g.2294G > C	g.86895A > G	g.86978G > A
g.2263A > G		D′ = 0.47	D′ = 0.97	D′ = 0.71
g.2294G > C	r^2^ = 0.05		D′ = 0.44	D′ = 1.00
g.86895A > G	r^2^ = 0.07	r^2^ = 0.07		D′ = 0.44
g.86978G > A	r^2^ = 0.05	r^2^ = 0.46	r^2^ = 0.03	
***SPP1* Loci**	g.50265G > A	g.50315C > T		
g.50265G > A		D′ = 0.78		
g.50315C > T	r^2^ = 0.22			

**Table 3 animals-13-00089-t003:** Genotype and allele frequency of SNPs of *ABCG2*, *CD44* and *SPP1* genes.

SNP Locus	Location	Number	Genotype	Genotype Frequency	Allele	Allele Number	Allele Frequency	χ^2^ HWE
*ABCG2*-g.57261A > G	Intron 1	77	AA	0.077	A	554	0.277	1.000
		400	AG	0.401	G	1444	0.723	
		522	GG	0.522				
*ABCG2*-g.80952G > T	Intron 1	162	GG	0.162	G	826	0.413	0.735
		502	GT	0.503	T	1172	0.587	
		335	TT	0.335				
*ABCG2*-g.94683A > G	Intron 5	256	AA	0.256	A	996	0.498	0.849
		484	AG	0.485	G	1002	0.502	
		259	GG	0.259				
*ABCG2-g.120017G > A*	Intron 13	272	AA	0.272	A	1040	0.521	0.994
		496	AG	0.497	G	958	0.479	
		231	GG	0.231				
*CD44*-g.2263A > G	Intron 2	660	AA	0.661	A	1617	0.809	0.722
		297	AG	0.297	G	381	0.191	
		42	GG	0.042				
*CD44*-g.2294G > C	Intron 2	250	CC	0.250	C	1007	0.504	0.955
		507	CG	0.508	G	991	0.496	
		242	GG	0.242				
*CD44*-g.86895A > G	exon 17	542	AA	0.543	A	1468	0.735	0.961
		384	AG	0.384	G	530	0.265	
		73	GG	0.073				
*CD44*-g.86978G > A	exon 17	464	AA	0.465	A	1356	0.679	0.93
		428	AG	0.428	G	642	0.321	
		107	GG	0.107				
*SPP1*-g. 50265G > A	Intron 1	44	AA	0.044	A	440	0.220	0.856
		352	AG	0.352	G	1558	0.780	
		603	GG	0.604				
*SPP1*-g. 50315 C > T	Intron 1	326	CC	0.326	C	1134	0.568	0.916
		482	CT	0.483	T	864	0.432	
		191	TT	0.191				

Note: SNP, single nucleotide polymorphisms; χ^2^ (HWE), Hardy–Weinberg equilibrium χ^2^ value.

**Table 4 animals-13-00089-t004:** Effects of different genotypes of SNPs of *ABCG2* gene on production performance of Holstein.

SNP Locus	Genotype	Record Number	Tested Day Milk Yield	Milk Fat Percentage	Milk Protein Percentage	Somatic Cell Score	Urea Nitrogen (mg/dL)
*ABCG2*-g.57261A > G	AA	1275	34.00 ± 0.20	3.87 ± 0.02 ^b^	3.28 ± 0.01	2.98 ± 0.04	13.83 ± 0.09
	AG	6729	34.68 ± 0.11	3.90 ± 0.01 ^ab^	3.28 ± 0.00	2.94 ± 0.02	13.90 ± 0.05
	GG	8990	34.45 ± 0.13	3.92 ± 0.01 ^a^	3.29 ± 0.01	2.88 ± 0.03	13.88 ± 0.06
	*p*		0.154	0.004 **	0.388	0.063	0.856
*ABCG2*-g.80952G > T	GG	2713	34.68 ± 0.20 ^a^	3.92 ± 0.02	3.28 ± 0.01 ^b^	2.82 ± 0.04 ^b^	13.84 ± 0.09 ^b^
	GT	8438	34.71 ± 0.11 ^a^	3.90 ± 0.01	3.28 ± 0.00 ^b^	2.91 ± 0.02 ^a^	13.86 ± 0.05 ^b^
	TT	5843	34.15 ± 0.13 ^b^	3.90 ± 0.01	3.29 ± 0.00 ^a^	2.95 ± 0.03 ^a^	13.94 ± 0.06 ^a^
	*p*		0.001 **	0.230	0.001 **	0.000 **	0.002 **
*ABCG2*-g.94683A > G	AA	4329	34.61 ± 0.15	3.91 ± 0.01	3.28 ± 0.01	2.92 ± 0.03	14.01 ± 0.07 ^a^
	AG	8059	34.47 ± 0.11	3.90 ± 0.01	3.29 ± 0.00	2.93 ± 0.02	13.81 ± 0.05 ^b^
	GG	4606	34.49 ± 0.15	3.91 ± 0.01	3.29 ± 0.01	2.87 ± 0.03	13.88 ± 0.07 ^ab^
	*p*		0.163	0.207	0.456	0.529	0.004 **
*ABCG2*-g.120017G > A	AA	4551	34.37 ± 0.15 ^b^	3.91 ± 0.01 ^a^	3.29 ± 0.01 ^a^	2.87 ± 0.03 ^b^	13.71 ± 0.07 ^b^
	AG	8419	34.77 ± 0.11 ^a^	3.91 ± 0.01 ^a^	3.28 ± 0.00 ^b^	2.95 ± 0.02 ^a^	14.01 ± 0.05 ^a^
	GG	4024	34.14 ± 0.16 ^b^	3.89 ± 0.01 ^b^	3.29 ± 0.01 ^a^	2.87 ± 0.03 ^b^	13.82 ± 0.07 ^b^
	*p*		0.007 **	0.041 *	0.001 **	0.005 **	0.000 **

Note: Data in the same column with different lowercase letters on the shoulder indicate significant differences (*p* < 0.05); * indicates that the differences reach a significant level (*p* < 0.05); ** indicates that the differences reach a highly significant level (*p* < 0.01).

**Table 5 animals-13-00089-t005:** Effects of different genotypes of SNPs of *CD44* gene on production performance of Holstein.

SNP Locus	Genotype	Record Number	Tested Day Milk Yield (kg)	Milk Fat Rate (%)	Milk Protein Rate (%)	Somatic Cell Score	Urea Nitrogen (mg/dL)
*CD44*-g.2263A > G	AA	11,107	34.64 ± 0.10	3.91 ± 0.01	3.28 ± 0.00	2.94 ± 0.02	14.03 ± 0.04 ^a^
	AG	5181	34.36 ± 0.14	3.90 ± 0.01	3.30 ± 0.01	2.88 ± 0.03	13.64 ± 0.06 ^b^
	GG	706	33.52 ± 0.40	3.97 ± 0.04	3.32 ± 0.01	2.77 ± 0.07	13.43 ± 0.17 ^b^
	*p*		0.081	0.354	0.090	0.052	0.000 **
*CD44*-g.2294G > C	CC	4404	34.24 ± 0.15	3.88 ± 0.01 ^b^	3.29 ± 0.01	2.98 ± 0.03 ^a^	13.59 ± 0.07 ^c^
	CG	8462	34.61 ± 0.11	3.92 ± 0.01 ^a^	3.28 ± 0.00	2.93 ± 0.02 ^a^	14.05 ± 0.05 ^a^
	GG	4128	34.59 ± 0.16	3.90 ± 0.02 ^ab^	3.29 ± 0.01	2.80 ± 0.03 ^b^	13.86 ± 0.07 ^b^
	*p*		0.114	0.010 *	0.109	0.001 **	0.000 **
*CD44*-g.86895A > G	AA	9266	34.37 ± 0.11 ^b^	3.91 ± 0.01 ^b^	3.30 ± 0.00 ^a^	2.88 ± 0.02	13.79 ± 0.05 ^b^
	AG	6538	34.51 ± 0.13 ^b^	3.89 ± 0.01 ^b^	3.28 ± 0.00 ^a^	2.95 ± 0.02	13.85 ± 0.06 ^b^
	GG	1190	35.61 ± 0.29 ^a^	3.97 ± 0.03 ^a^	3.24 ± 0.01 ^b^	2.94 ± 0.06	14.82 ± 0.13 ^a^
	*p*		0.003 **	0.007 **	0.000 **	0.127	0.000 **
*CD44*-g.86978G > A	AA	8087	34.45 ± 0.11	3.93 ± 0.01 ^a^	3.29 ± 0.00	2.87 ± 0.02	13.92 ± 0.05
	AG	7177	34.56 ± 0.12	3.90 ± 0.01 ^ab^	3.29 ± 0.00	2.93 ± 0.02	13.88 ± 0.05
	GG	1730	34.61 ± 0.25	3.86 ± 0.02 ^b^	3.27 ± 0.01	3.04 ± 0.05	13.75 ± 0.11
	*p*		0.252	0.004 **	0.097	0.316	0.060

Note: Data in the same column with different lowercase letters on the shoulder indicate significant differences (*p* < 0.05); * indicates that the differences reach a significant level (*p* < 0.05); ** indicates that the differences reach a highly significant level (*p* < 0.01).

**Table 6 animals-13-00089-t006:** Effects of different genotypes of SNPs of *SPP1* gene on production performance of Holstein.

SNP Locus	Genotype	Record Number	Tested Day Milk Yield	Milk Fat Rate	Milk Protein Rate	Somatic Cell Score	Urea Nitrogen
*SPP1*-g. 50265G > A	AA	745	33.61 ± 0.37 ^b^	3.89 ± 0.03	3.29 ± 0.01	2.90 ± 0.07	13.23 ± 0.16 ^b^
	AG	6121	34.47 ± 0.13 ^a^	3.90 ± 0.01	3.29 ± 0.00	2.93 ± 0.02	13.78 ± 0.06 ^a^
	GG	10,128	34.60 ± 0.10 ^a^	3.92 ± 0.01	3.28 ± 0.00	2.90 ± 0.02	13.99 ± 0.04 ^a^
	*p*		0.016 *	0.215	0.619	0.843	0.000 **
*SPP1*-g. 50315 C > T	CC	5427	34.50 ± 0.13	3.88 ± 0.01 ^b^	3.28 ± 0.00	2.89 ± 0.03	14.02 ± 0.06
	CT	8272	34.50 ± 0.11	3.92 ± 0.01 ^a^	3.29 ± 0.00	2.93 ± 0.02	13.81 ± 0.05
	TT	3295	34.54 ± 0.18	3.91 ± 0.01 ^a^	3.28 ± 0.01	2.90 ± 0.03	13.83 ± 0.08
	*p*		0.420	0.016 *	0.360	0.396	0.134

Note: Data in the same column with different lowercase letters on the shoulder indicate significant differences (*p* < 0.05); * indicates that the differences reach a significant level (*p* < 0.05); ** indicates that the differences reach a highly significant level (*p* < 0.01).

## Data Availability

The data presented in this study are available in this article.

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
