# Peer review of "Molecular Marker-Assisted Selection of ABCG2, CD44, SPP1 Genes Contribute to Milk Production Traits of Chinese Holstein"

_animals, 2022, doi:10.3390/ani13010089_

Round 1
Reviewer 1 Report
I reviewed the article entitled Molecular Marker-assisted Selection of ABCG2, CD44, SPP1 Genes Contribute to Production Performance of Chinese Holstein. The authors performed an analysis of SNPs for three genes linked with milk production performance of Chinese Holstein cows with the use of MassARRAY MALDI-TOF System technology. The following issues should be corrected prior to the publication:
1. In this experiment, hair follicle samples from 999 cows in four farms were collected from 8580 cows that passed the quality control. Could you explain why such number of the samples was collected and what quality control was performed in order to select these 999 cows?
2. All the gene names throughout the whole manuscript should be written in italic.
3. There is a typo in the title and in some other parragraphs
4. The description of the collection of biological material should be presented as a list of activities that have been performed, and not as an instruction on how to do it. Please rewrite parragraphs [71-74].
5. The quality of Sequencing peak map could be improved.
Author Response
Dear Reviewer,
We sincerely appreciate the careful reading of our manuscript entitled “Molecular Marker-assisted Selection of ABCG2, CD44, SPP1 Genes Contribute to Production Performance of Chinese Holstein” (ID: animals-2076742) and your valuable suggestions. Our point-by-point responses to your comments have been provided below, along with a clear indication of the location of the relevant revisions.
- Thank you for your comments. These 999 cows have completed pedigree information (more than three generations) and at least two complete DHI records (more than six records in a single lactation period); To approximately follow a normal distribution, SCC values were converted to somatic cell score (SCS) values through the calculation of (log2 (SCC/100,000) +3). The following quality control measures were conducted on these test-day records: 1) records were kept of cows ranging from parity 1 to parity 3; 2) records were kept for milk production ranging from 5 kg to 80 kg, for DIM stages between 5 and 305, and for SCS records ranging from 0 to 9; 4) records were removed if recorded less than 6 times in a parity; and 5) records were removed for cows with first-calving ages less than 22 months or more than 36 months.
- Thank you for your comments. I have already rewritten all the gene names in italic at the revised manuscript.
- Thank you for your comments. I have already checked and revised some typo mistakes.
- Thank you for your comments. I have carefully considered the part of the collection of biological material at “Sample collection”, found it wordy and deleted this description at the revised manuscript.
- I have already rewritten the part of the collection of biological material at “Sample collection”.
- Thank you for your comments. I have already replaced the new sequencing peak map.

Reviewer 2 Report
1. “Production Performance” and “Hol-stein” in the title are not accurately written.
2. The layout of Figures 2 to 4 is not good. It is suggested that the same mutation site be arranged vertically.
3. The method description of functional annotation and signal pathway analysis is lacking, especially the significant difference test method, which is critical to the accuracy of the results.
4. In the model of association analysis, the lack of age and paternity effects can lead to inaccurate results. This is main defect.
5. The results of Table 4 and Figure 6, Table 5 and Figure 7, Table 6 and Figure 8 are duplicate. it is recommended to delete one of them.
6. As mentioned in the article, linkage disequilibrium exists between some SNPs. Therefore, the results of haplotype and its association analysis are more convincing and more helpful for molecular selection of cows.
In summary, the lack of age and paternity effects in the model of association analysis leads to inaccurate results. Therefore, the manuscript is not suitable for publication in the current version.
Author Response
Dear Reviewer,
We sincerely appreciate the careful reading of our manuscript entitled “Molecular Marker-assisted Selection of ABCG2, CD44, SPP1 Genes Contribute to Production Performance of Chinese Holstein” (ID: animals-2076742) and your valuable suggestions. Our point-by-point responses to your comments have been provided below, along with a clear indication of the location of the relevant revisions.
- Thank you for your comments. After consideration, we have revised the title to “Molecular Marker-assisted Selection of ABCG2, CD44, SPP1 Genes Contribute to Milk Production Traits of Chinese Holstein”.
- Thank you for your comments. We have revised the Figure 2 to 4, and arranged the same mutation site vertically in the revised manuscript.
- Thank you for your comments. We have listed the online websites for the functional annotation and signal pathway analysis on “Materials and Methods”. We used SPSS (Ver. 18.0) software to analyze the significant differences. And we have added some description at the revised manuscript.
- Thank you for your comments. Because these experimental cows were of the same age (24-month) and the same parity (parity 2), these were no age and paternity effects listed in the model of association analysis. We have added some description on the linear model in the revised manuscript.
- Thank you for your comments. Some Tables and Figures represent the same result, we considered there were specific error values in these Tables, and more intuitive to see the difference among different genotypes in these Figures. We will delete one of them if necessary.
- Thank you for your comments. Our study indicated that combined haplotypes have no differences on milk production traits of Chinese Holstein. According previous study (Posada and Crandall, 2001), the high-frequency haplotypes have probably been present in the population for a long time. Consequently, most new mutants are derived from common haplotypes, implying that rarer variants represent more recent mutations and are more likely to be related to common haplotypes than to other rare variants.
Posada, D., Crandall, K.A., 2001. Intraspecific gene genealogies: trees grafting into networks. Trends Ecol. Evol. 16, 37-45.

Round 2
Reviewer 1 Report
No further comments on the manuscript. In my opinion, the manuscript could be published in this form.
Author Response
Dear Reviewer,
We sincerely appreciate the careful reading of our manuscript entitled “Molecular Marker-assisted Selection of ABCG2, CD44, SPP1 Genes Contribute to Milk Production Traits of Chinese Holstein” (ID: animals-2076742) and your valuable suggestions. Thank you for your previous comments again.
Best regards,
Yujia Sun
Reviewer 2 Report
In the author's reply, they proposed: "Because these experimental cows were of the same age (24-month) and the same parity (parity 2), these were no age and paternity effects listed in the model of association analysis." Age and parity are very important for the accuracy or reliability of all results. So, the statement of " these experimental cows were of the same age (24-month) and the same parity (parity 2) is must be included in the “2.1. Sample collection” or "2.6 Statistical analysis". Otherwise, the article cannot be published.
Author Response
Dear Reviewer,
Thank you for your valuable suggestions. We have added the description about age and parity in the “2.1 Sample collection” in the revised manuscript. Thank you for all your comments again.
Best regards,
Yujia Sun